# A Paleo-Perspective of 21st Century Drought in the Hron River (Slovakia)

Igor Leščešen [1], Abel Andrés Ramírez Molina [2] and Glenn Tootle [3,*]

1   Institute of Hydrology, Slovak Academy of Sciences, Dúbravská Cesta 9, 841 04 Bratislava, Slovakia; lescesen@uh.savba.sk
2   Department of Computer Science, The University of Alabama, Tuscaloosa, AL 35487, USA; aaramirez@crimson.ua.edu
3   Department of Civil, Construction and Environmental Engineering, The University of Alabama, Tuscaloosa, AL 35487, USA
*   Correspondence: gatootle@eng.ua.edu; Tel.: +1-307-399-6666

**Abstract**

The Hron River is a vital waterway in central Slovakia. In evaluating observed streamflow records for the past ~90 years, the Hron River displayed historically low hydrologic summer (April–May–June–July–August–September (AMJJAS)) streamflow for the 10-, 20-, and 30-year periods ending in 2020. When using self-calibrated Palmer Drought Severity Index (scPDSI) proxies developed from tree-ring records, skillful regression-based reconstructions of AMJJAS streamflow were developed for two gauges (Banská Bystrica and Brehy) on the Hron River. The recent observed droughts were compared to these reconstructions and revealed the Hron River experienced extreme drought in the 21st century. A further comparison of observed wet (pluvial) periods revealed that the most extreme robust streamflow periods in the observed record were frequently exceeded in the reconstructed (paleo) record. The Hron River has recently been experiencing decline, and we hypothesize that this decline may be associated with anthropogenic influences, the natural climatic cycle, or the changing climate.

**Keywords:** drought; streamflow; dendrohydrology; Slovakia

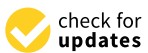

## 1. Introduction

Recent findings from the latest Intergovernmental Panel on Climate Change (IPCC) report underscore that climate change has already inflicted substantial damage on terrestrial and freshwater ecosystems and is projected to exacerbate climate-related risks across both short and long timescales [1]. The increasing frequency and severity of climatic extremes have already contributed to shortages in food and water systems [1], and they pose a significant threat to future water availability [2,3].

However, this study does not aim to predict future droughts, instead, it reconstructs past hydrological variability. Understanding historical changes in river flow is vital for contextualizing current variability and improving resilience to future extremes. Central Europe, as a transitional zone between Mediterranean and continental precipitation regimes, exhibits a complex landscape climate interplay, making it particularly sensitive to moisture shifts and drought stress [4–6]. Indeed, paleohydrological reconstructions from tree rings have revealed multi-centennial drought episodes in Central Europe, such as the work by Büntgen et al. [7] and Dobrovolný et al. [8], highlighting the importance of long-term perspectives in hydrological science.

Tree-ring proxies have proven to be a valuable tool for reconstructing past hydro-climatic conditions, and they provide insights into the variability of river flows from centennial to millennial time scales [9,10]. Despite significant progress in the reconstruction of tree-ring data, studies remain focused on a limited number of European river catchments. The majority of published reconstructions have been carried out in adjacent regions, including the recent efforts by Obertelli [11] in the Rhine and Po catchments. Their study showed mixed success with relatively high reconstruction capability in the Rhine Basin but poor performance in the Po Basin, likely due to the influence of alpine runoff [11,12]. A more comprehensive study by Nasreen et al. [12] reconstructed the runoff of fourteen European watersheds in the period 1500–2000 AD and found extreme drought years in 1540, 1669, and 1921. However, many Central European catchments, including the Upper Danube and its tributaries, have not yet been adequately studied in this context.

Traditional tree ring-based reconstructions usually use direct relationships between tree growth indices and river flow variability [9,10]. However, novel approaches have recently been developed that utilize tree ring-derived hydroclimatic indices as proxies for streamflow, broadening the applicability of dendrohydrology at regional and continental scales. A notable example is the Old-World Drought Atlas (OWDA), which provides a high-resolution, annually resolved self-calibrated Palmer Drought Severity Index (scPDSI) derived from 106 tree-ring chronologies covering 5414 grid points across Europe from 0 to 2012 AD [13]. Similar to previous studies [11,12], in this study, we leverage OWDA-derived scPDSI data as a proxy for streamflow. For this purpose, we identify the relevant grid cells within a 450 km radius of the study basin's centroid, a methodology successfully applied in previous work [9].

While previous research in Central Europe has extensively examined paleoclimate records for variables such as temperature and hydroclimatic variability [14–17], dedicated streamflow reconstructions remain relatively rare. Recent studies have successfully reconstructed long-term streamflow variability in major European river systems, including the Danube and Sava Rivers. A 250-year reconstruction of the Danube River near Tulcea, Romania, utilized oak (*Quercus* sp.) tree-ring chronologies spanning 1728–2020 AD, demonstrating strong correlations between tree-ring indices and November–July streamflow [3]. Similarly, a 500-year reconstruction of the Danube near Orsova, Romania, incorporated OWDA-derived scPDSI proxies, yielding significant correlations with seasonal streamflow variability [13]. The most comprehensive effort in the region to date has focused on the Sava River in Slovenia, where the Stepwise Linear Regression (SLR) method was applied to tree ring-based scPDSI proxies, successfully identifying multi-decadal periods of both low- and high-flow conditions [16]. Additionally, Trlin et al. [18] reconstructed summer streamflow in the Sava River near Jasenovac, Croatia, using narrow-leaved ash (Fraxinus angustifolia) tree-ring chronologies and identifying significant correlations with May–August discharge. Notably, scPDSI-based reconstructions exhibited even stronger correlations than traditional tree-ring chronologies, reinforcing the utility of hydroclimatic proxies for streamflow reconstruction [16,18]. The paleo-reconstruction of Sava River discharge in Serbia has shown that the 2000–2022 period ranks as the lowest 23-year flow interval in the observed record and one of the driest over the past ~500 years [19].

This study therefore not only fills a critical gap by reconstructing seasonal April–May–June–July–August–September (AMJJAS) streamflow in the Hron River Basin, a region not previously investigated using OWDA-derived scPDSI proxies, but it also introduces an ensemble, multi-window SLR approach to rigorously quantify reconstruction uncertainty. By comparing reconstructed hydrological extremes with instrumental records, regional reconstructions, and historical drought documentation, we establish the regional coherence of past drought regimes and place recent low-flow events into a long-term, paleo

context. Ultimately, our findings provide a novel, century-to-millennial scale benchmark that enhances drought risk assessment and water resource planning in Central Europe.

## 2. Study Area

The Hron River catchment is located in central and southern Slovakia (Figure 1), originating in the Low Tatras Mountains at an altitude of 934 m and flowing 279 km to its confluence with the Danube River near Štúrovo at 112 m above sea level [20]. As Slovakia's second-longest river, its catchment covers approximately 5465 km$^2$, about 11% of the country's territory [21,22]. The geography of the Upper Hron region is critical to its hydrology, with extensive forest cover (approx. 65%) in mountainous areas like the Poľana and Veporské vrchy ranges [4]. This forested, high-altitude terrain directly influences local climate and the availability of natural archives for streamflow reconstruction, such as tree rings. The region's climate is strongly governed by this topography. Climatic conditions during the April to September growing season are particularly relevant to drought formation. This period is characterized by the year's warmest temperatures, with average July temperatures ranging between 14 °C and 16 °C [23]. Furthermore, a substantial portion of the annual precipitation falls during these late spring and summer months, making this period's climate balance crucial for both forest health and streamflow levels [24]. Despite its importance to Slovakia's drainage system, the Hron faces significant hydrological challenges, primarily intensifying droughts driven by climate variability in the 21st century [4].

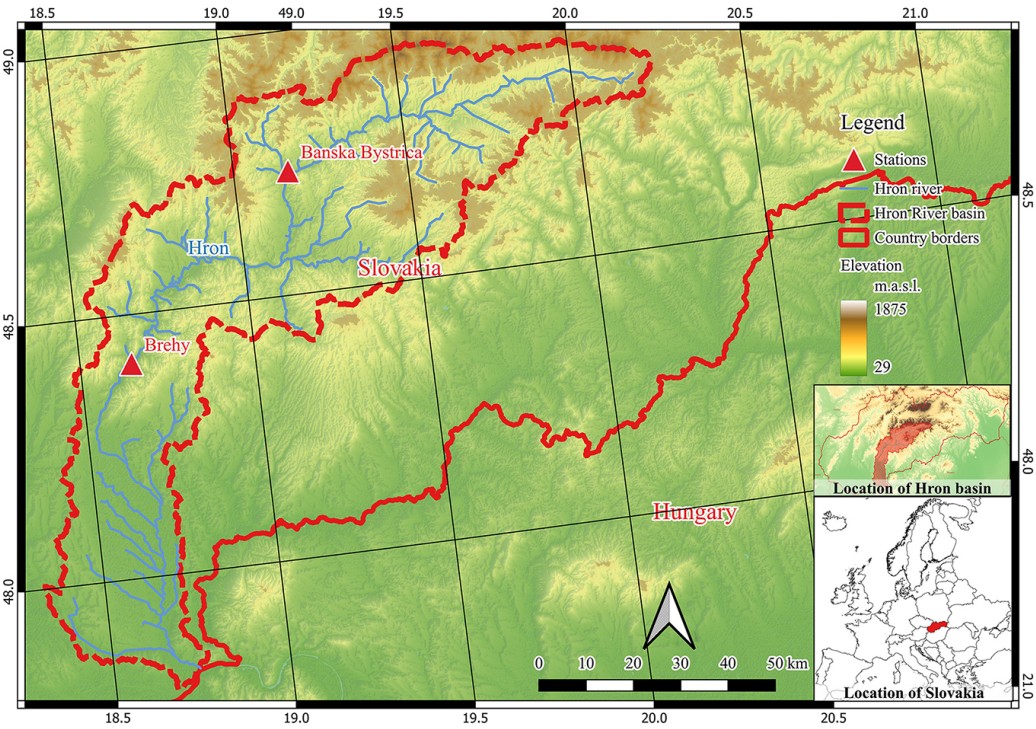

**Figure 1.** Geographical position of the research basin and the stations used for this study.

For this study, instrumental streamflow data from two gauging stations on the Hron River, Banská Bystrica, and Brehy (Figure 1), were utilized. The data were provided by the Slovak Hydro-meteorological Institute (SHMI) and consist of the mean monthly discharge records for the 90-year period spanning from 1931 to 2020. The long-term mean annual discharge is 25.7 m$^3$/s at the Banská Bystrica station and 46.2 m$^3$/s at the Brehy station. The seasonal cycle for both stations is characterized by the highest discharge in April, following snowmelt, and the lowest in September (Figure 2).

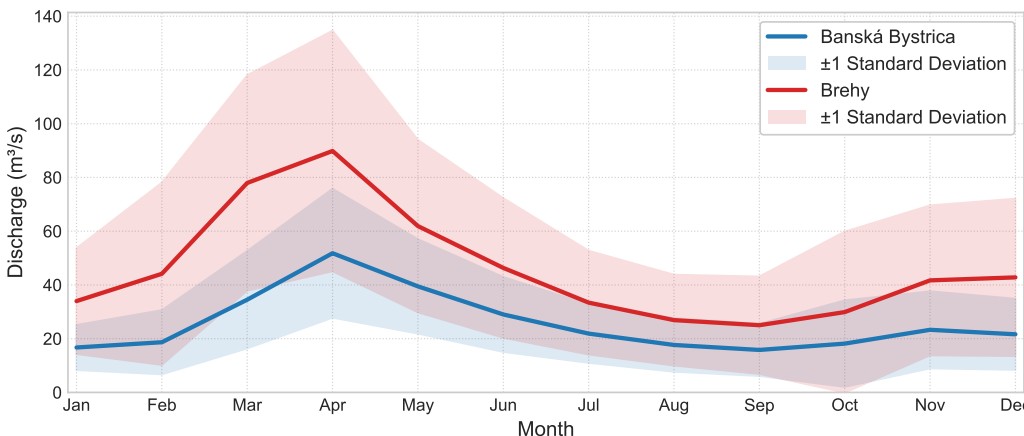

**Figure 2.** Mean monthly discharge for the Banská Bystrica and Brehy stations over the 1931–2020 period. The solid lines represent the monthly mean, while the shaded areas represent ±1 standard deviation around the mean.

## 3. Data and Methods

This paper uses a 90-year average monthly flow database (1931–2020) of the two selected stations on the Hron River (Banská Bystrica and Brehy). This database is considered to be sufficient because it generally requires a database of at least 50 years to distinguish between variability and trends [25]. This data set was supplied by the Slovak Hydrometeorological Institute (SHMI). The visual review found that there were no observation gaps in the selected river stations' data. The data quality control measures were strictly implemented by SHMI, which was responsible for collecting and maintaining water data. The institute follows strict standards to ensure the accuracy and consistency of the data series used in this study. They followed not only their internal instructions, but also the recommendations of the World Meteorological Organization (WMO). The monthly flowrates ($m^3$/s) were converted to monthly volume (Million-Cubic-Meters (MCM)), and we selected the hydrological summer (April–May–June–July–August–September (AMJJAS)) for the season of interest in developing the reconstruction of the Hron River–Banská Bystrica streamflow gauge and the Hron River–Brehy streamflow gauge.

The Old-World Drought Atlas (OWDA) provides 5414 grid points across Europe, using tree ring-derived (0 to 2012 AD) annual (June–July–August (JJA)) self-calibrated Palmer Drought Severity Index (scPDSI) [13]. Building on the work of Ho et al. [9,10], who demonstrated for U.S. streams that a summer (JJA) drought index can be the most skillful predictor for reconstructing broader seasonal and annual streamflow, we applied the JJA OWDA scPDSI to reconstruct AMJJAS seasonal streamflow using an Stepwise Linear Regression (SLR) model (Forward–Backwards) per Woodhouse [26]. While the Hron's peak discharge occurs in spring from snowmelt, the JJA scPDSI proxy is effective because tree growth integrates the full warm season water balance, including moisture from both winter snowpack and summer precipitation, which collectively drive AMJJAS streamflow. To capture uncertainty in the development of our AMJJAS reconstruction, we developed multiple SLR models by examining 30-year, 40-year, and 50-year windows within the 90-year observed period on record. To mitigate the risk of overfitting inherent to SLR models with numerous potential predictors, a multi-step validation protocol including stringent predictor pre-screening, temporal stability analysis, and cross-validation was employed [16,27,28]. To be considered in the SLR model, retained scPDSI cells must meet or exceed a positive correlation with AMJJAS seasonal streamflow equivalent to 99% ($p \leq 0.01$). Additionally, the scPDSI cells retained (e.g., 99% positive significance) were evaluated for stability by performing a moving correlation window, using one-third of

the record, and all correlation values must be positive. A variety of skill statistics were evaluated, including the coefficient of variation ($R^2$) (which must exceed 0.50); $R^2$-predicted ($R_P^2$) (calculated using drop-one cross-validation–while this method can underestimate predictive uncertainty in serially correlated data, it was selected as a standard approach that maximizes the data available for calibration, which is critical for the relatively short windows used in this study–and must be within 0.1 of $R^2$); Variation Inflation Factor (VIF), (which determines whether multicollinearity is present and must be less than 10 with a value of 1 being optimal) [29]; Durbin–Watson (DW) (determines whether auto-correlation was present) [30]; and Sign Test (ST) (compares instrumental and reconstructed flows to see if the model is over- or underestimating). No two models could contain the same regression equation (i.e., same scPDSI cells) to avoid weighting the multi-model AMJJAS average. The reconstructed annual AMJJAS flows were bias-corrected using a quantile mapping approach [31]. The observed 10-year, 20-year, and 30-year maximums (pluvials) and minimums (droughts) were compared to the reconstructed multi-model streamflow for the 10-year, 20-year, and 30-year filters (running averages).

## 4. Results and Discussion

### 4.1. Streamflow Reconstruction and Paleo-Drought Analysis

In evaluating the observed April–May–June–July–August–September (AMJJAS) streamflow, the most recent year (end-year 2020) in the period on record represented the lowest multi-year (10-year, 20-year, and 30-year) period multiple times. For the Banská Bystrica gauge, the lowest 10-year period was from 2011 to 2020 (369 Million-Cubic-Meters (MCM)), the lowest 20-year period was from 2001 to 2020 (398 MCM), and the lowest 30-year period was from 1991 to 2020 (406 MCM). For the Brehy gauge, the lowest 10-year period was from 2011 to 2020 (592 MCM), the lowest 20-year period was from 1990 to 2009 (626 MCM), and the lowest 30-year period was from 1991 to 2020 (648 MCM). Thus, the recent (21st century) observed flows were generally the lowest in the ~90-year observed period on record.

For the Banská Bystrica gauge, we identified eight skillful reconstructions of AMJJAS streamflow (Table 1), and for the Brehy gauge, we identified three skillful reconstructions of AMJJAS streamflow (Q) (Table 2).

**Table 1.** Hron River (Banská Bystrica) April–May–June–July–August–September Streamflow (AMJJAS Q), Reconstruction Period, coefficient of variation ($R^2$), $R^2$-predicted ($R_P^2$), Variation Inflation Factor (VIF), Durbin–Watson (DW), Sign Test (ST), and Regression Equation for Regression Model (self-calibrated Palmer Drought Severity Index (scPDSI) proxy cell number in parenthesis). Reconstruction windows (50-year period—Black; 40-year period—Red; and 30-year period—Blue).

| Period | $r^2$ | $R_P^2$ | VIF | DW | ST | Equation |
|--------|-------|---------|-----|-----|-------|----------|
| 1963–2012 | 0.51 | 0.45 | 1.3 | 2.3 | 24/26 | AMJJAS Q = 499.7 + 52.4 (2194) + 33.0 (1780) |
| 1967–1996 | 0.60 | 0.52 | 1.1 | 1.8 | 14/16 | AMJJAS Q = 490.0 + 51.6 (1944) + 26.8 (2545) |
| 1967–2006 | 0.50 | 0.42 | 1.1 | 2.1 | 20/20 | AMJJAS Q = 502.6 + 46.8 (2104) + 25.0 (1886) |
| 1970–2009 | 0.57 | 0.47 | 1.2 | 1.7 | 19/21 | AMJJAS Q = 480.4 + 25.4 (2196) + 25.8 (2000) + 24.3 (2376) |
| 1971–2010 | 0.55 | 0.50 | 1.1 | 2.0 | 18/22 | AMJJAS Q = 461.6 + 43.0 (2237) + 21.3 (1839) |
| 1972–2011 | 0.59 | 0.50 | 1.6 | 2.0 | 19/21 | AMJJAS Q = 466.9 + 31.8 (2237) + 19.2 (1839) + 19.2 (2000) |
| 1973–2012 | 0.62 | 0.52 | 1.4 | 1.6 | 20/20 | AMJJAS Q = 478.6 + 29.2 (2104) + 25.3 (2000) + 16.3 (2376) |
| 1983–2012 | 0.66 | 0.59 | 1.7 | 1.7 | 17/13 | AMJJAS Q = 509.4 + 44.1 (2194) + 40.3 (1943) |

**Table 2.** Hron River (Brehy) April–May–June–July–August–September Streamflow (AMJJAS Q), Reconstruction Period, coefficient of variation ($R^2$), $R^2$-predicted ($R_p^2$), Variation Inflation Factor (VIF), Durbin–Watson (DW), Sign Test (ST), and Regression Equation for Regression Model (self-calibrated Palmer Drought Severity Index (scPDSI) proxy cell number in parenthesis). Reconstruction windows (40-year period—Red; 30-year period—Blue).

| Period | $R^2$ | $R_p^2$ | VIF | DW | ST | Equation |
|--------|-------|---------|-----|----|----|----------|
| 1966–1995 | 0.52 | 0.43 | 1.1 | 2.0 | 14/16 | AMJJAS Q = 785.8 + 78.7 (2000) + 46.7 (2006) |
| 1970–2009 | 0.54 | 0.46 | 1.0 | 1.9 | 23/17 | AMJJAS Q = 774.0 + 78.0 (1727) + 51.8 (2006) |
| 1974–2003 | 0.51 | 0.42 | 1.1 | 1.8 | 14/16 | AMJJAS Q = 734.4 + 65.3 (1727) + 42.3 (1839) |

We combined the eight April–May–June–July–August–September (AMJJAS) bias-corrected reconstruction models for the Banská Bystrica gauge and the three AMJJAS bias-corrected reconstruction models for the Brehy gauge to capture uncertainty (gray area) and applied 10-year, 20-year, and 30-year filters (black line is the average of the eight or three models) (Figures 3 and 4). The observed multi-year lows (droughts) are shown as a horizontal reference line (red line), and a visual inspection was conducted of the ~2000-year reconstruction.

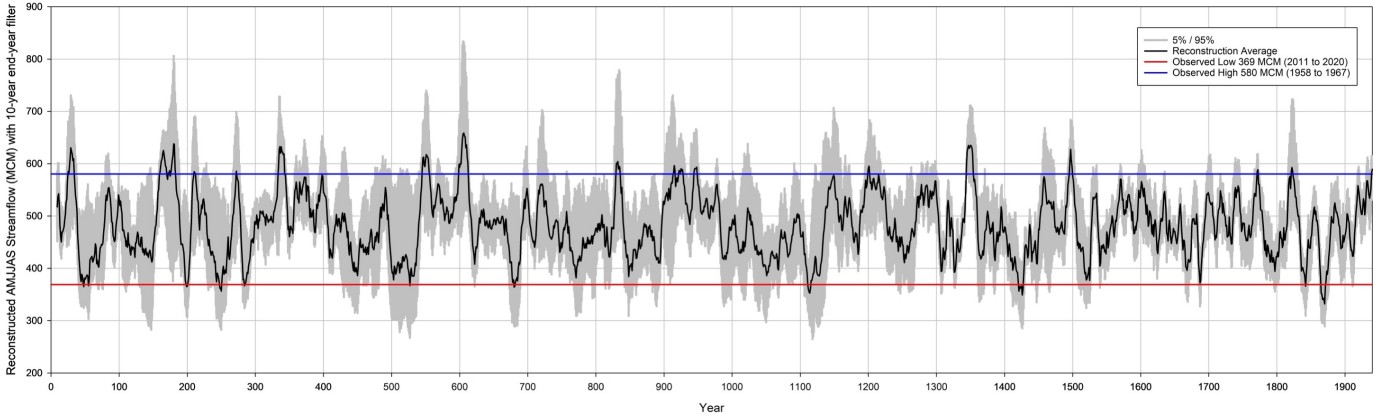

(**a**) 10-year end-year filter.

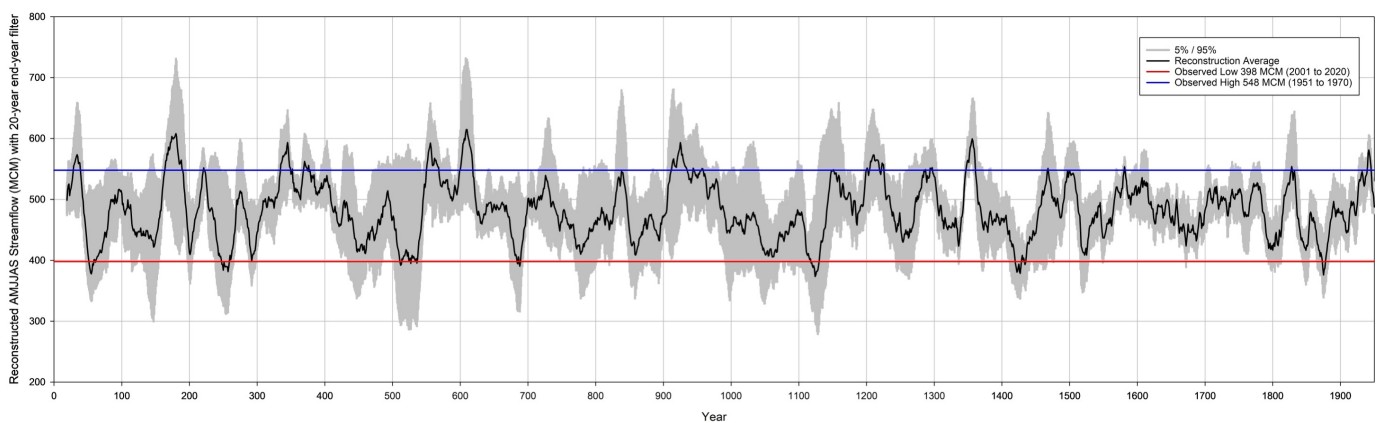

(**b**) 20-year end-year filter.

**Figure 3.** *Cont.*

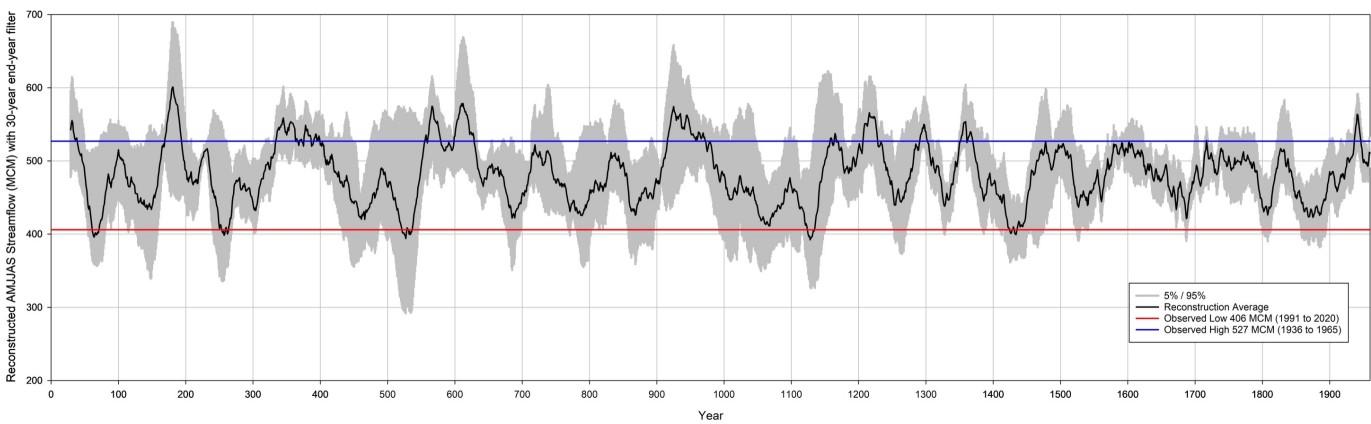

(**c**) 30-year end-year filter.

**Figure 3.** Banská Bystrica gauge bias-corrected reconstructed AMJJAS Q with (**a**) 10-year (**b**) 20-year and (**c**) 30-year end-year filters. The gray area represents the 5th/95th percentile when combining the eight reconstruction models, while the black line represents the average of the eight models. The horizontal red line and blue line represent, respectively, the most extreme multi-year low (drought) and high (pluvial) flows from the 1931–2020 observed period, plotted for comparison against the full paleo-reconstruction.

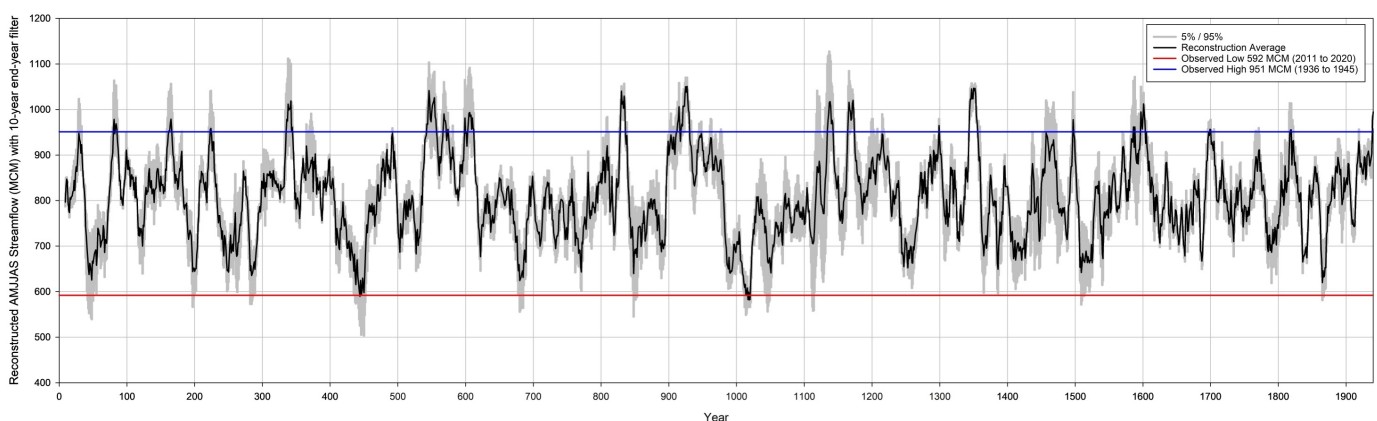

(**a**) 10-year end-year filter.

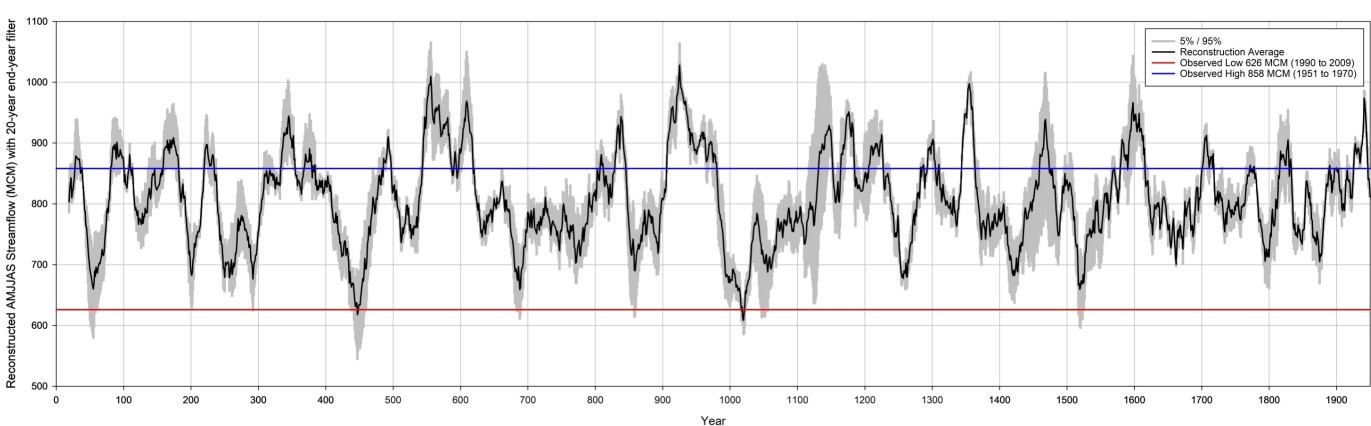

(**b**) 20-year end-year filter.

**Figure 4.** *Cont*.

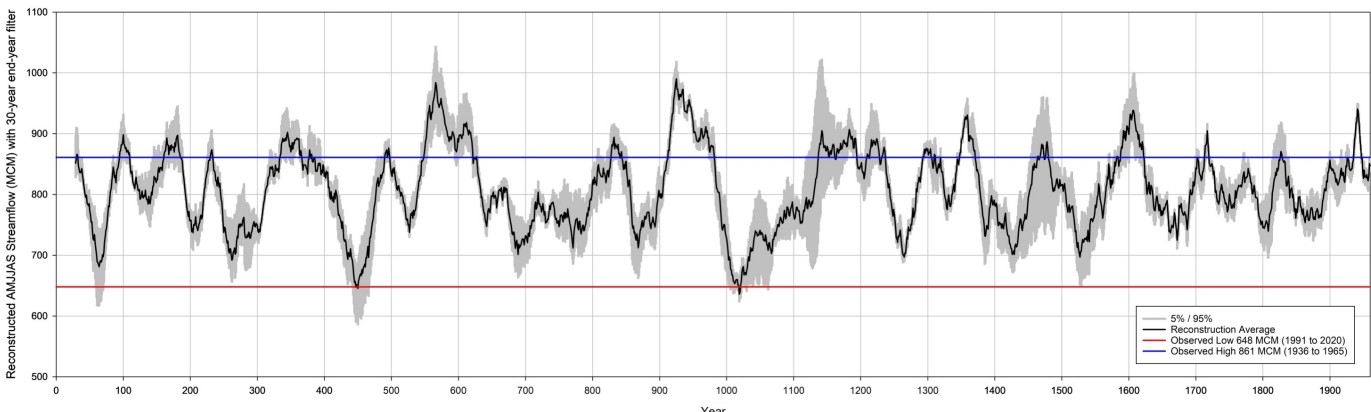

(**c**) 30-year end-year filter.

**Figure 4.** Brehy gauge bias-corrected reconstructed AMJJAS Q with (**a**) 10-year (**b**) 20-year (**c**) 30-year end-year filters. The gray area represents the 5th/95th percentile when combining the eight reconstruction models while the black line represents the average of the eight models. The horizontal red line and blue line represent, respectively, the most extreme multi-year low (drought) and high (pluvial) flows from the 1931–2020 observed period, plotted for comparison against the full paleo-reconstruction.

A visual examination of the reconstructed series highlights that streamflow has markedly declined during the 21st century, positioning recent observed droughts among the most severe in the ~2000-year reconstruction. The severity of the modern period is underscored by the fact that reconstructed (paleo) droughts only infrequently exceeded the most extreme droughts of the instrumental record. For the upstream Banská Bystrica gauge, such exceptional paleo-droughts occurred during multi-decadal events in the 1st, 3rd, 6th, 12th, and 15th centuries (Figure 3). For the downstream Brehy gauge, which integrates a larger catchment area, this occurred primarily during the 5th and 11th centuries (Figure 4). However, the persistence of declining flows since 1991 suggests a trend toward intensifying drought conditions, positioning recent droughts among the most severe of the last five centuries and highlighting their increasing rarity in a long-term context. Similar declining flows have also been reported in other parts of Europe, including the Danube River [32], Sava River [33], Lauter River, and Rhine River [34].

A more detailed assessment across both gauges and all filters confirms this rarity; for each case, reconstructed (paleo) droughts only occasionally surpassed the severity of the most extreme observed droughts. Specifically, at the Banská Bystrica gauge, the worst observed 10-year drought was exceeded approximately 10 times in the paleo record, the 20-year drought about 8 times, and the 30-year drought about 25 times (Figure 3). For the Brehy gauge, the most severe observed 10-, 20-, and 30-year droughts were each surpassed only about two times in the paleo record (Figure 4).

In stark contrast, the paleo record for pluvial (wet) periods tells a different story. The most robust streamflows in the observed record were frequently and substantially exceeded in the paleo-reconstruction at both gauges. This divergence—where modern droughts are extreme but modern wet periods are not—suggests a shift toward a drier hydroclimatic regime in the 21st century.

Additional historical evidence from documentary sources further supports the severity of past droughts in Central Europe. In the Czech lands, five outstanding drought events (1540, 1590, 1616, 1718, and 1719) have been identified from records dating back to 1090 CE [35]. The 1540 drought is widely regarded as one of the most extreme events in terms of precipitation deficit and excessive warmth, surpassing many droughts observed in the last century [13]. These historical droughts align with tree ring-based reconstructions

from the Old-World Drought Atlas (OWDA), reinforcing the occurrence of past multi-year dry periods that match or exceed the severity of modern droughts. However, the increasing persistence of low flows in the 21st century suggests that recent drought conditions may be approaching the severity and duration of the most extreme multi-decadal droughts of the past 1500 years.

It is a well-established principle in river science that the cumulative impact of human activities, such as water withdrawals for agriculture and industry and alterations to river channels, tends to increase in a downstream direction as population density and land use intensity grow [36]. Applying this principle to the Hron River, we hypothesize that the upstream Banská Bystrica gauge reflects hydrological conditions with fewer anthropogenic disturbances compared to the downstream Brehy gauge. The Brehy station, being further down the catchment, likely integrates a greater degree of these pressures, which could potentially amplify the recorded severity of low-flow periods in the instrumental record. This may explain why we observed fewer paleo-droughts exceeding the worst observed droughts at the Brehy gauge.

Paleo-droughts recorded on the Hron River closely correspond to those documented in the southern Carpathian Basin, particularly on the Sava River, where significant drought events occurred in 55 CE, 250 CE, 450 CE, 687 CE, 1067 CE, 1422 CE, 1522 CE, and 1861 CE [19]. These events were also identified in the paleo record of the Hron River, indicating a broader regional coherence in historical drought patterns. An additional visual inspection reveals the most robust events (highest flows or pluvial periods) in the observed records for both gauges were repeatedly exceeded in the paleo record (Figures 3 and 4). The wettest observed periods were frequently exceeded in the paleo record, again revealing how dry the 21st century has been so far.

Our reconstruction shows strong coherence with large-scale continental patterns, which have been established through multi-proxy synthesis studies [37]. However, it is important to contextualize these findings by discussing regional differences. For instance, while our results align well with other lowland and mid-altitude European reconstructions, they may differ in the timing of specific drought events when compared to studies from the high Alps, such as the Adige River Basin [28]. Such discrepancies can often be attributed to differing hydrological drivers; high-altitude Alpine catchments are heavily influenced by snow melt dynamics, which dictate a distinct seasonality of flow, whereas the Hron Basin's flow is more directly tied to seasonal precipitation and evapotranspiration. Despite these regional variations driven by local catchment characteristics, the widespread nature of major reconstructed drought periods suggests that the large-scale climatic forces behind them were widespread and lasting [13].

### 4.2. Discussion of Model Performance and Climate Drivers

A closer evaluation of the reconstruction models retained show a stronger (more skillful) relationship between scPDSI (i.e., tree-ring proxies) and AMJJAS Q in more recent records (Figure 5). When (solely) evaluating $R^2$ for each reconstruction model generated (30-year, 40-year, and 50-year reconstruction windows), $R^2$ generally increases when the overlapping AMJJAS Q and scPDSI records are more recent. We hypothesize that this could be the result of more accurate streamflow records in recent years due to improved collection and monitoring mechanisms [38,39].

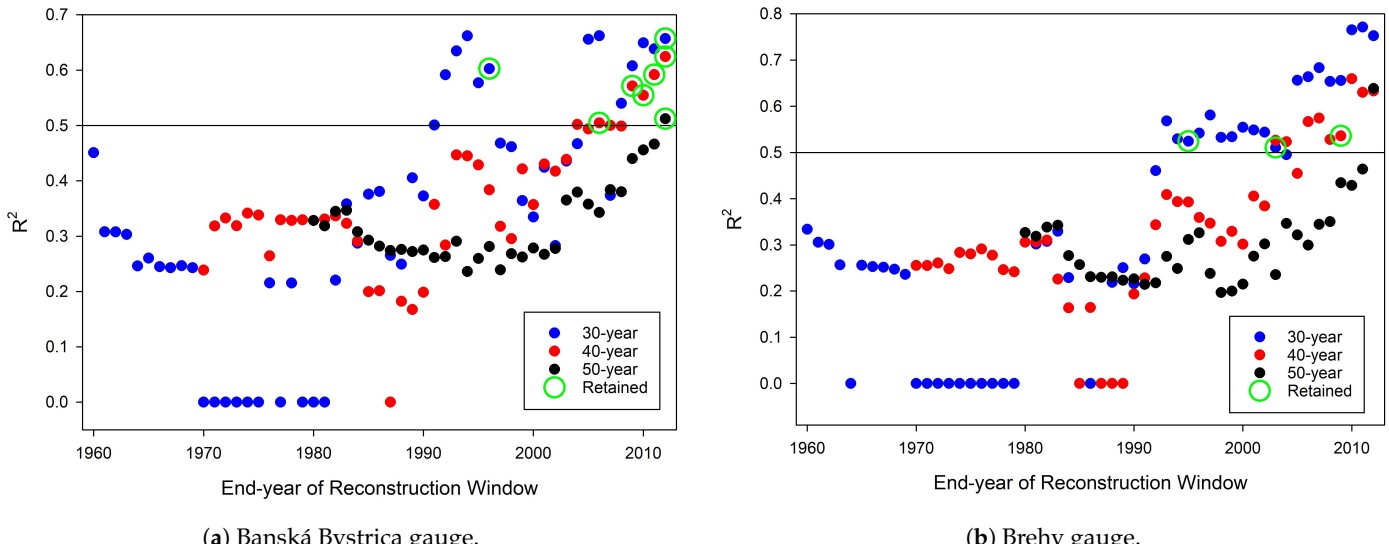

(**a**) Banská Bystrica gauge.　　　　　　　　　　(**b**) Brehy gauge.

**Figure 5.** Reconstruction model $R^2$ for (**a**) Banská Bystrica gauge and (**b**) Brehy gauge for varying (50-year period—Black, 40-year period—Red, and 30-year period—Blue) reconstruction windows. The retained models are identified with a green circle.

This observed decline in the Hron River's streamflow during the 21st century, established by the fact that multi-year AMJJAS flows are the lowest in the ~90-year period on record, can be attributed to a combination of factors, including climate change, anthropogenic activities, and natural variability. Trend analyses indicate that drought is a recurring climate feature in the region, yet there is a clear shift toward more arid conditions [4,40]. Recent decades have witnessed rising temperatures, particularly in the summer and early autumn, accompanied by a decline in precipitation relative to the 1961–1990 long-term average. This trend is consistent with the findings of Fendeková et al. [41], who documented reduced runoff during warmer months, as well as those of Sleziak et al. [42], who observed decreasing summer runoff and increasing winter runoff in the Nitra catchment, linked to rising air temperatures and shifting precipitation patterns. Similarly, Minárik et al. [43] reported a significant increase in mean annual temperature and precipitation over the past three decades, reinforcing concerns about hydrological shifts. Moreover, measurements from 2005 to 2019 confirm a continuous rise in both mean monthly and annual temperatures [44], supporting the notion that higher temperatures enhance evapotranspiration, thereby reducing runoff despite occasional precipitation increases. This pattern has been observed in multiple Slovak river basins, including the Krupinica, Litava, and Ipeľ [45] Basins, as well as in Czech river basins, where declining water availability from March to October has been documented [46]. Regardless of the specific drivers, the paleo record demonstrates that the early 21st century represents one of the driest periods in the past ~2000 years, underscoring the severity of recent hydrological changes.

## 5. Conclusions

This study reconstructs the variability of streamflows in the Hron River Basin over the past century using tree-based proxy networks, in particular, the self-calibrated Palmer Drought Severity Index (scPDSI), based on the Old-World Drought Atlas (OWDA). Our findings show that the period 2000–2020 is one of the driest multi-decade intervals in the last 500 years, which highlights recent exceptional hydrological changes. The reconstruction records reveal that extreme droughts and pluvial conditions are changing, indicating an increase in "whiplash" hydrological events that may signal changes in regional climate dynamics. By applying a Stepwise Linear Regression (SLR) model to the moisture climate

index, we expanded the instrument discharge record and improved our understanding of the historical extreme of the watershed. We use rigorous statistical frameworks, including cross-validation techniques, bias correction methods, and multi-level evaluation criteria to ensure the reliability of the reconstructed flow-current series. These results strengthen the potential of dendrohydrology for palaeohydrological assessment and demonstrate its value in climate impact studies and water resource planning. The reconstruction underscores the unprecedented severity of recent low water conditions in the Hron catchment, which are consistent with broader trends in Central Europe. The results suggest that anthropogenic climate change is exacerbating hydrological extremes, posing a challenge to future water availability and ecosystem resilience. The expanded dataset on streamflow is an important resource for water managers, policy makers, and researchers, and it provides a historical benchmark for assessing future hydrological risks. Future research should aim to refine the paleohydrologic reconstructions by integrating additional proxy datasets, such as lake sediments and speleothem isotopes, to improve spatial and temporal resolution. Furthermore, machine learning approaches such as random forests and neural networks could improve prediction accuracy and capture complex hydroclimatic relationships to support adaptive water management strategies in the face of climate change.

**Author Contributions:** Conceptualization, G.T. and I.L.; methodology, G.T.; software, A.A.R.M.; validation, I.L., A.A.R.M., and G.T.; formal analysis, I.L., A.A.R.M., and G.T.; investigation, G.T.; resources, I.L.; data curation, I.L.; writing—original draft preparation, I.L. and G.T.; writing—review and editing, I.L., A.A.R.M., and G.T.; visualization, I.L., A.A.R.M., and G.T.; supervision, G.T.; project administration, I.L., A.A.R.M., and G.T.; funding acquisition, G.T. All authors have read and agreed to the published version of the manuscript.

**Funding:** This research was supported by the project 09I03-03-V04-00186 "Streamflow Drought Through Time," funded by the European Union's Next Generation EU as part of the Recovery and Resilience Plan of the Slovak Republic, specifically through the initiative for more efficient management and enhancement of funding for research, development, and innovation. Additional support was provided by the National Science Foundation Research Traineeship (NSF NRT), Water: Research to Operations (Water: R2O) (2152140) at The University of Alabama, and by the National Oceanic and Atmospheric Administration (NOAA), awarded to CIROH through the NOAA Cooperative Agreement with The University of Alabama, NA22NWS4320003.

**Data Availability Statement:** The data supporting the results of this study are available and archived at https://alabama.box.com/s/e0z1bf5lburxeqkfggw1z91aljafy2jk, accessed on 27 February 2025.

**Acknowledgments:** The authors wish to thank The University of Alabama, Alabama Water Institute (AWI), and the Cooperative Institute for Research to Operations in Hydrology (CIROH) for their institutional support. The statements, findings, conclusions, and recommendations are those of the authors and do not necessarily reflect the views of NOAA.

**Conflicts of Interest:** The authors declare no conflicts of interest.

## Abbreviations

The following abbreviations are used in this manuscript:

| | |
|---|---|
| AMJJAS | April–May–June–July–August–September |
| AMJJAS Q | April–May–June–July–August–September Streamflow |
| DW | Durbin–Watson |
| IPCC | Intergovernmental Panel on Climate Change |
| JJA | June–July–August |
| MCM | Million-Cubic-Meters |
| OWDA | Old-World Drought Atlas |
| Q | streamflow |

| $R^2$ | coefficient of variation |
|---|---|
| $R^2_p$ | $R^2$-predicted |
| scPDSI | self-calibrated Palmer Drought Severity Index |
| SHMI | Slovak Hydro-meteorological Institute |
| SLR | Stepwise Linear Regression |
| ST | Sign Test |
| VIF | Variation Inflation Factor |
| WMO | World Meteorological Organization |

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
