# Peer review of "A Paleo-Perspective of 21st Century Drought in the Hron River (Slovakia)"

_hydrology, doi:10.3390/hydrology12070169_

Round 1
Reviewer 1 Report
Comments and Suggestions for Authors
Introduction section:
Lines 15-20: First, the research context is too broad and lacks relevance. The introduction opens with a reference to the IPCC report and emphasizes the global impacts of climate change, but does not directly relate to the issue of drought. Second, the focus of the research context needs to be adjusted. The core of the paper is the reconstruction of historical drought changes in the Hron River from paleo-climate data, but the current context emphasizes too much future climate change, which makes readers misinterpret it as a 'future prediction' study rather than a 'paleo climate reconstruction' study.
Lines 21-23: The logical structure of this sentence needs to be improved, as the direct cut from global climate change to 'hydrological research in Central Europe' seems too abrupt. It is recommended to add more information about the importance of the study area, such as the hydrological significance of the Central European region and the Hron River, as well as current threats such as droughts, in order to enhance the context and coherence of the article.
Lines 70-73: The objectives of the study are vaguely stated. Statements such as “This study aims to contribute to the growing body of research on dendrohydrology”, while having a macroscopic significance, do not clarify the specific research questions and their scientific innovations.
Lines 73-76: Recommended for deletion.
Study area section:
Figure 1: Consider adding latitude and longitude grids to clarify the spatial orientation of the study area.
Lines 78-93: Suggest streamlining out irrelevant climate data descriptions. For example, if the paper discusses drought conditions from April to September, the description of winter temperatures is redundant. It is recommended to focus on geographic elements that may influence tree species and drought.
Results and Discussion section:
“4. Results and Discussion” section: it is suggested to add (1) (2)... subsection headings to improve the readability and structure of the paper.
Lines 186-188: Please provide references to support the idea and further discuss anthropogenic impacts on drought in the watershed.
Lines 216-218: Please provide references to support the hypothesis.
Lines 200-211: The first sentence summarizes the previous paragraph and should not appear in this paragraph. This paragraph discusses the differences with other studies. It is suggested that additional relevant literature be added to elaborate on exactly what the differences are and why they exist.
Reviewer 2 Report
Comments and Suggestions for Authors
The manuscript titled “A Paleo Perspective of 21st Century Drought in the Hron River (Slovakia)” uses self-calibrated Palmer Drought Severity Index (scPDSI) proxies derived from tree-ring records and applies Stepwise Linear Regression (SLR) to reconstruct April–May–June–July–August–September (AMJJAS) streamflow at two gauges (Banská Bystrica and Brehy) on the Hron River. By comparing these reconstructions with observed streamflow records, the author concludes that the Hron River has experienced extreme drought conditions in the 21st century. The manuscript is generally well-structured and clearly written. However, I have several major concerns outlined below:
- There is a temporal mismatch in using June–July–August (JJA) scPDSI to reconstruct AMJJAS streamflow. The authors should provide a clear justification for this choice in the “Data and Methods” section.
- While SLR is commonly used, it is prone to overfitting, particularly with a large number of candidate predictors. It is unclear whether any steps were taken to reduce this risk. The authors should elaborate on any overfitting mitigation strategies.
- The use of drop-one cross-validation may underestimate predictive uncertainty in time series data due to temporal autocorrelation. The authors should discuss whether alternative or temporally aware validation approaches were considered.
- Additional details should be provided regarding the bias correction method applied to the reconstructed streamflow.
- The y-axes of Figures 3 and 4 appear to end around ~1950, yet some key results and interpretations (e.g., Lines 152 and 178) refer to events in the late-20th and 21st centuries, which are not visible on these figures.
Please also refer to my moderate and minor comments in the attached PDF.

Round 2
Reviewer 1 Report
Comments and Suggestions for Authors
No
Author Response
We sincerely thank Reviewer 1 for their positive evaluation and support for the publication of our manuscript. We appreciate your time and effort in reviewing our work.
Reviewer 2 Report
Comments and Suggestions for Authors
Thank you for your efforts in addressing my comments.
Please see my remaining comments below:
Regarding your response to Major Comment #1: It doesn’t appear that Ho et al. (2016, 2017) specifically tested multiple temporal aggregations (e.g., spring, autumn, full water-year PDSI) to conclude that a summer drought index is the most skillful predictor of streamflow. Rather, their work relied on summer-only reconstructions (i.e., LBDA scPDSI). Additionally, since the Hron River typically experiences its highest discharge in April due to snowmelt (Lines 104-106), I am still confused why JJA scPDSI—which reflects summer drought conditions—would be an effective predictor of AMJJAS streamflow. I would expect that winter precipitation and spring temperature would play more important roles in influencing streamflow.
Additionally, the legend for “Elevation” should be retained in Figure 1.
Round 3
Reviewer 2 Report
Comments and Suggestions for Authors
Thank you for your efforts in addressing all of my comments.